# Spatial Turnover and Functional Redundancy in the Ants of Urban Fragments of Tropical Dry Forest

**Lina María Ramos Ortega \*** and **Roberto J. Guerrero**

Facultad de Ciencias Básicas, Universidad del Magdalena, Santa Marta 470003, Colombia; rguerrero@unimagdalena.edu.co
\* Correspondence: linaramosmo@unimagdalena.edu.co

**Abstract:** Spatial and temporal variation in the diversity of ants in four urban fragments of the tropical dry forest in the city of Santa Marta was evaluated. The fragments were sampled four times in the dry and rainy season, from October 2019 to January 2020, using pitfall traps, mini-Winkler bags, baits, and manual collection. Both alpha and beta taxonomic diversity and their components were quantified. The functional groups were established based on proposals for Neotropical ant species. A total of 7 subfamilies, 37 genera, and 84 species were collected. Richness varied spatially from 33 to 61 species, but between the two seasons it was 72 and 76 species. Sites N01 and N02 had greater diversity than N03 and N04. In all the fragments, soil ants were dominated by *Ectatomma ruidum*, but litter ants showed a structure with less dominant species. The dissimilarity between fragments was 60–80%, attributable mainly to turnover (50–70%) but not to nestedness (10%). Seventeen functional groups were identified. Taxonomic diversity of ants in urban fragments in Santa Marta showed marked spatial variation, without influence from the seasons. Despite taxonomic turnover, there was broad similarity in functional groups between the fragments, indicating ecological equivalence of species between the ant assemblages.

**Keywords:** beta diversity partitioning; ecological species equivalence; *Ectatomma ruidum*; sampling coverage; spatial differentiation

## 1. Introduction

The tropical dry forest (TDF) is a terrestrial ecosystem distributed in lowlands between 0 and 1000 m. a. s. l, with a marked rainfall seasonality (precipitation less than 100 mm) and several months of drought [1]. The biota that inhabit the TDF are adapted to water-stress conditions, making it a unique ecosystem due to its high levels of endemism [2]. In Colombia, the TDF is one of the most threatened ecosystems and presents a high degree of degradation and fragmentation derived largely from the expansion of agriculture and livestock, as well as the growth of infrastructure due to human settlements [3], currently only 8% of its original extension within six large regions [3]. The Colombian Caribbean presents the largest extension of TDF (417,838 ha) with the highest proportion in the Cesar, Bolívar, and Magdalena departments [2]. In the latter, the largest extensions of the TDF are distributed within the Tayrona National Park, but with some remnants in the peri-urban area and within the city of Santa Marta.

The city of Santa Marta has experienced population and urban growth without further planning, which has generated the fragmentation of extensive areas of natural forests connected to the plant formations of the Sierra Nevada de Santa Marta. Currently, urban forest fragments worldwide provide important environmental services, such as biodiversity protection, $CO_2$ capture and storage, seed dispersal, water regulation, and landscape improvement. This situation imposes the need to develop studies that describe and evaluate the biodiversity and ecosystem services of environments, such as the remnants of the TDF that persist in the city of Santa Marta.

The TDF in the Santa Marta region faces the same challenges of threatening the ecosystem on a national scale, including the little research that has been carried out in these environments, which has mainly contemplated vegetation, mammals, birds, amphibians, and dung beetles [3]. However, a common group in dry forests are ants, which possess special ecological characteristics (e.g., seed dispersers) to be used as indicators of biodiversity, disturbance, and restoration of ecosystems [4].

In the Neotropical region, mainly Brazil, ants in TDF areas have been studied to evaluate the seasonal and successional effects of vegetation on the ant community structure [5–7]. Likewise, in Mexico, researchers have studied the patterns of ant diversity along anthropogenic disturbance gradients [8,9]. In Colombia, TDF studies on ants are represented mainly in the western and southwestern region, addressing biodiversity issues [10–13]. In the Caribbean, there are few studies on TDF ants, which have been oriented towards the generation of inventories and diversity assessments [14–16], and poorly towards the analysis of the community and functional structure of ants in disturbed environments [17,18]. In the Santa Marta region, information is even more limited, and the scope of the research has been mainly taxonomic [19,20].

Knowledge about the TDF is still insufficient, even for charismatic groups such as plants and birds [21]. These authors have suggested, considering the little information available, that patterns of beta diversity for groups, such as plants, birds, and bats, of the TDF in Colombia vary according to the spatial scale considered, for which it is important and necessary to study the ecological patterns and processes at local and regional scales due to the singularities that each of these fragments or remnants can harbor.

Here, we evaluate the spatial and temporal variation in the taxonomic diversity and functional groups of ants in urban fragments of the tropical dry forest in the city of Santa Marta. This information contributes to the knowledge needs of the TDF both in Colombia and at the local level and becomes an input for its conservation, as well as for decision making regarding management and planning of the urban environment.

## 2. Materials and Methods

### 2.1. Study Area

The city of Santa Marta, in northern Colombia, has a total area of 2393.35 km$^2$, of which 55.10 km$^2$ corresponds to the urban area [22]. The average annual temperature is 27 °C, the average annual rainfall is 608.8 mm, and there is a unimodal rainfall regime [23]. The predominant vegetation cover in these areas is characterized by forests and scrublands with an isomegathermal floor (high and constant temperatures throughout the year), with pronounced xeromorphic characters due to the longer dry season, reaching nine months a year [24,25]. According to the physiognomic and floristic characteristics, these plant formations are defined as those of a tropical subxerophytic zonobiome [24,25].

The sampling locations were as follows (Figure 1): 1. Dry Forest Plot University of Magdalena (N01), an urban fragment of dry forest with an extension of 2.8 ha, located on the campus of the University of Magdalena (11°13′ N 74°11′ W, 21 m. a. s. l). It has been in the process of passive restoration for around 7 years and is subject to conservation, research, and environmental education; 2. Fragment of Dry Forest Quinta de San Pedro Alejandrino (N02), located within the urban area of Santa Marta (11°13′ N 74°10′ W, 26 m. s. n. m), with a total extension of 22 ha, surrounded by buildings and green areas, which are subject to conservation, research, and environmental education; 3. The Green Iguana Reserve (N03), which is in the rural area of the city of Santa Marta, El Mosquito (11°10′ N 74°10′ W, 100 m. s. n. m). It corresponds to private property, with a total extension of 19 ha. This reserve is not subject to cultivation pressures or any extractive activity, although there are some human settlements in its surroundings; 4. Quebrada Seca (N04), which is in the rural area of the city of Santa Marta (11°13′ N 74°08′ W, 81 m. s. n. m). Quebrada Seca corresponds to private property with an approximate extension of 6 ha, which seems not to be subject to cultivation pressures or any extractive activity, although there are some human settlements in its surroundings.

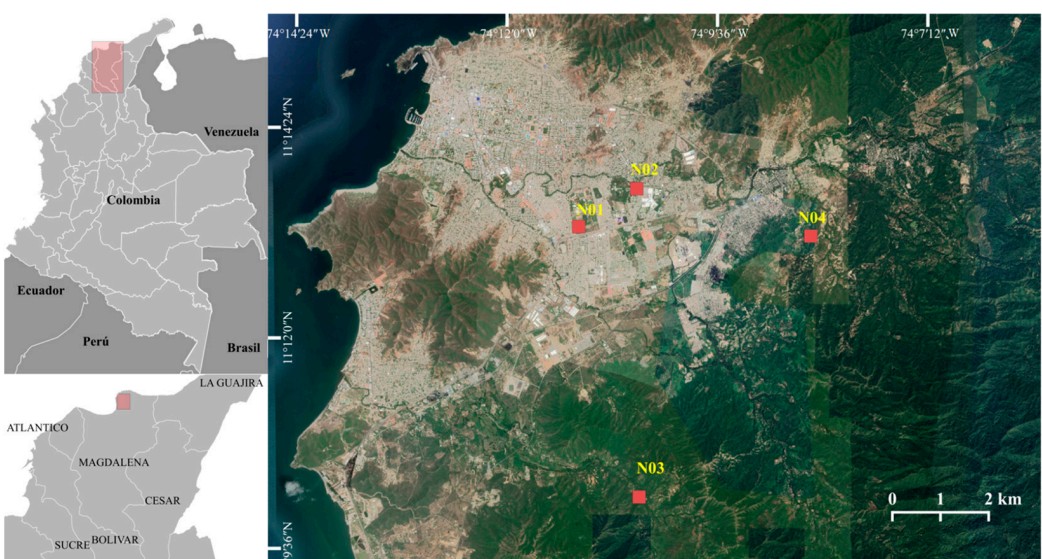

**Figure 1.** Map indicating the four fragments of tropical dry forest studied in the city of Santa Marta, Colombia.

*2.2. Ants Sampling*

Four samplings were carried out between the months of October and December 2019 and January 2020, trying to cover the months of high and low precipitation, respectively. The sites N01, N02, and N03 were sampled in all the months considered. In the case of N04, there were records from the month of November. The choice of collection methods and the number of sampling units was made based on the recommendations established in [26].

To collect the ants from the soil, three 100 m linear transects were delimited in each site, in which 10 pitfall traps (207 mL container) were placed 10 m apart, filled to 2/3 of the volume with water and concentrated ethanol (96%), before acting for 48 h. Also, 15 protein baits (tuna) and 15 carbohydrate baits (biscuit mix with condensed milk) were placed in the epigeal stratum; after 30 min, they were checked and collected with the ants present. The ants associated with the litter were collected in 5–10 quadrats of 1 m$^2$ of litter, arranged randomly within the transects; the litter sample was held within a mini-Winkler bag for 48 h to extract the ants present in the litter. To complement the sampling, the ants foraging on the arboreal vegetation within the area delimited by the transects were collected manually for one hour.

For the taxonomic identification of the ants, the specialized keys in [27] were used. The specimens were deposited in the Biological Collections of the University of Magdalena-CBUMAG (RNC N°207).

*2.3. Data Analysis*

Diversity was evaluated as the effective number of species ($^qD$), where the exponent q determines the influence of species abundance on diversity values and varies from zero to infinity [28]. Three q values were used: order 0 ($^0D$, species richness); 1 ($^1D$, effective number of common species); and 2 ($^2D$, effective number of dominant species) [28]. The variation in the assemblage structure was analyzed, based on the shape of the range–abundance curves for ants on the ground and litter. The relative abundance of the ant species was measured as the capture frequency for each of the species, considering the arrangement of the number of pitfall traps and litter samples in each site.

Estimates and comparisons of diversity were made between the different assemblages under the same or similar sampling coverage (Ĉm), which is the value that indicates the proportion of the statistical population that is represented by the species captured [29–31]. Ĉm takes values from 0 (minimum completeness) to 1 (maximum completeness). For incidence-based diversity comparisons, 95% confidence intervals (CIs) were used, and differences

were determined following the recommendations of [32], where non-overlapping among CIs indicates significant differences. The sampling coverage values, the diversity of order q and their CIs were calculated using the iNEXT package for R [33].

To establish the trends in the spatial and temporal variation in the ant assemblage (ground and litter), non-metric multidimensional scaling (nMDS) analysis was used based on the Bray–Curtis similarity index (only "pitfall" and Winkler data), using the capture frequency as a measure of abundance. To determine statistical differences between sites and sampling times, ANOSIM similarity analysis was performed. The calculations were developed in the PRIMER v6 program [34].

The beta diversity ($\beta_{cc}$) was estimated using the approximation of [35], dividing its components into species turnover ($\beta_{-3}$) and richness differences ($\beta_{rich}$); in the latter, it refers to the absolute difference in the number of species that each site contains [35]. The Jaccard dissimilarity index was used to calculate the beta diversity values and its components, using the BAT package for R [36].

The trends in the spatial change of the functional groups in ants were evaluated, which were established considering the proposal of [37] for Neotropical ants. In addition, information about the biology of groups, such as fungus growers [38] and specialized soil predators/foragers [39], was used.

## 3. Results

### 3.1. Composition and Completeness of Sampling

A total of 30,250 individuals grouped into 7 subfamilies, 37 genera, and 84 species (Appendix A) was collected. The subfamilies Myrmicinae (59.5%), Ponerinae (11.9%), Formicinae (10.7%), and Pseudomyrmecinae (7.1%) contributed about 90% of the ant richness. The genera *Pheidole*, *Cephalotes*, *Crematogaster*, *Pseudomyrmex*, *Camponotus*, *Solenopsis*, and *Strumigenys* contributed more than 50% of the recorded species.

*Ectatomma ruidum* (Roger, 1860) was the species with the highest frequency of capture (81%), followed by a group of eight species with frequencies between 10 and 24%, such as *Pheidole fallax* Mayr, 1870, *Acromyrmex santschii* (Forel, 1912), and *Solenopsis bicolor* (Emery, 1906) (Appendix A). About 24 species were in the 2–10% range (e.g., *Pheidole inversa* Forel, 1901, *Temnothorax subditivus* (Wheeler, 1903), and *Odontomachus bauri* Emery, 1892. A total of 51 species were rare, with frequencies less than 2% (Appendix A).

Sampling coverage for pitfall traps and baits ranged between 93 and 98%. In the case of the samples collected with mini-Winkler bags, the coverage had variations between 80 and 95%, with the lowest values present during the dry season. In the case of soil ants, the sampling deficit was less than 7%, while for ants associated with leaf litter, it varied between 20 and 5%.

### 3.2. Alpha Diversity

In general, the pitfall traps and mini-Winkler methods collected the highest number of species (67 and 60, respectively), followed by hand collection (46 species) and baits (33 species) (Appendix A). The number of species recorded with baits was between two and four times less than that collected with pitfall traps, and at all sites, the species collected with baits were represented or contained in the pitfall samples. Based on this, the variation in the soil ants was analyzed using only the information from the pitfall traps.

The richness of ants remained constant between both seasons, of which there were 49 species for those that live in the leaf litter (mini-Winkler bags) and 60 for those that forage on the ground (pitfall traps); however, the number of individuals varied substantially between seasons and between sampling methods (2497 individuals during the rainy season and 1072 during the dry season for those collected with mini-Winkler bags compared to 8529 individuals during the rainy season and 15,767 during the dry season were collected with pitfall traps). The richness between sites ranged from 33 to 61 species, with the highest richness in N01 and N03, 56 species in N02, and the lowest value in N04. Between seasons,

the richness values were relatively similar with 72 species in the rainy season and 76 in the dry season.

For ants that forage on the ground, greater diversity was observed in sites N01 and N02 with respect to N03 and N04, which was significant only for N04 (Figure 2a); this trend was repeated in both climatic seasons and for the 1D and 2D diversity orders (Figure 2b,c). Litter ants showed variations in diversity like those of the soil but without significant differences between seasons (Figure 2). Similar values for the three orders of diversity were also recovered, except in N04, which registered the lowest values in the study and was significantly different from the other sites.

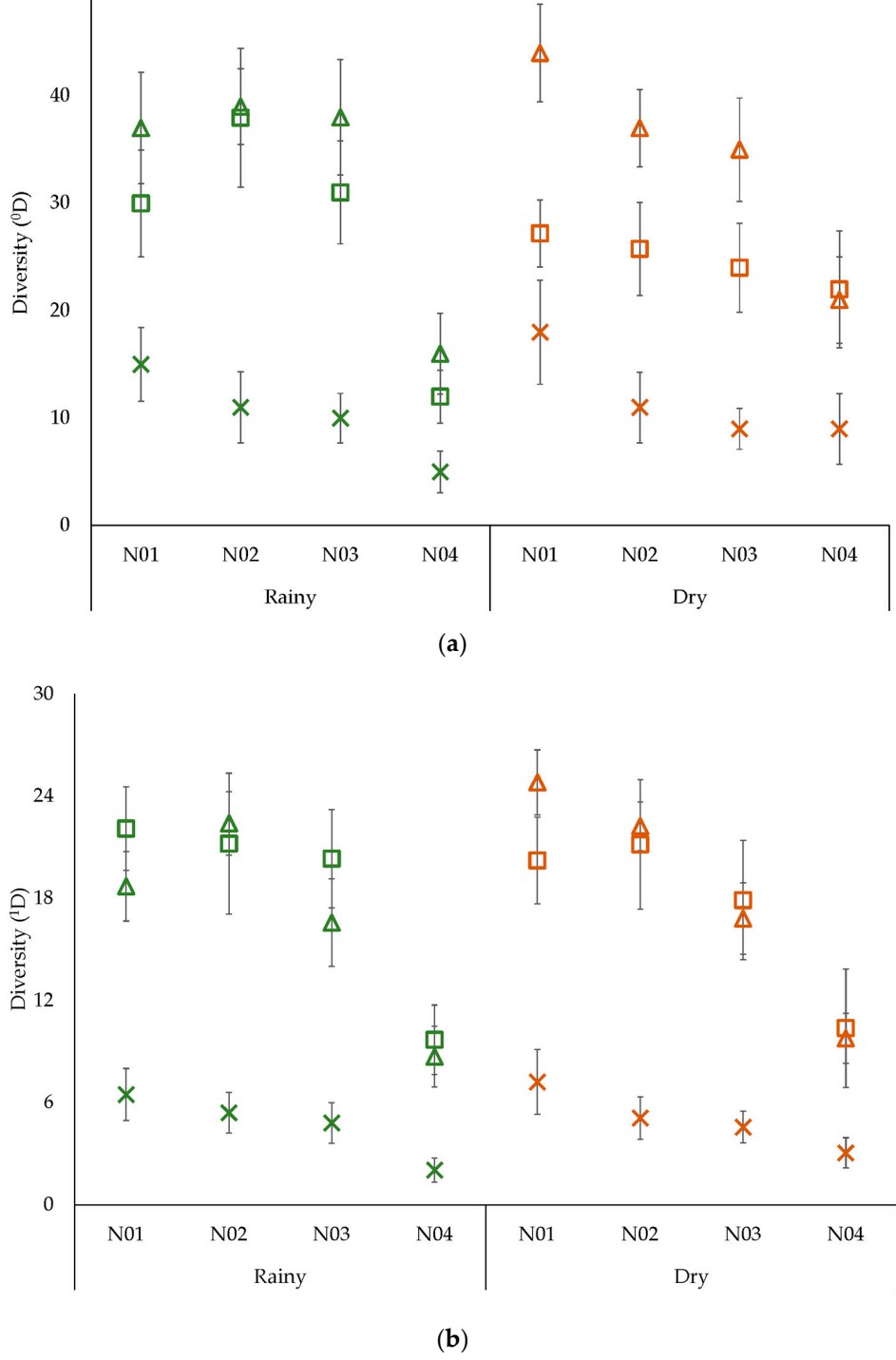

**Figure 2.** *Cont.*

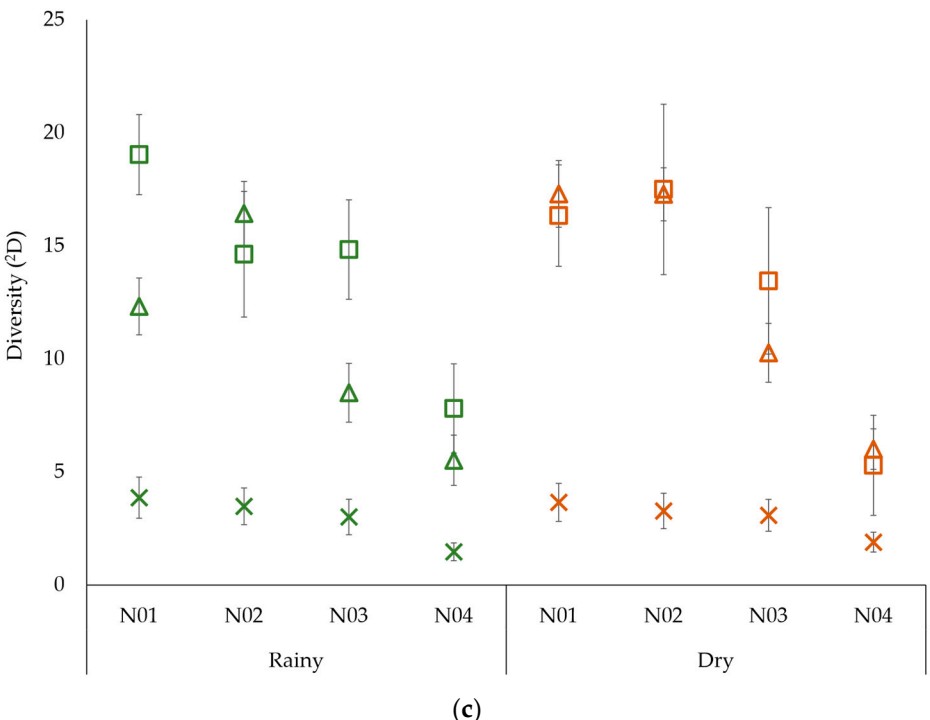

**(c)**

**Figure 2.** Diversity expressed as the effective number of ant species (qD) in urban fragments of TDF in the city of Santa Marta. (**a**) Richness; (**b**) common species; (**c**) dominant species. The triangles correspond to the pitfall traps; the circles correspond to the ants extracted from the leaf litter with mini-Winkler bags; and the Xs correspond to the ants attracted to the baits. The bars indicate the confidence intervals (CIs) of each of the measurements.

The rank–abundance curves show that a few species concentrated a higher fraction of the individuals in the assemblage (Figure 3a,b). Soil ants were strongly dominated by *Ectatomma ruidum* in all sites, presenting a relative abundance between 94 and 98% (Figure 3a). Other species with moderate to high contribution corresponded to *Acromyrmex santschii*, *Pheidole fallax*, *Pheidole guajirana* Wilson, 2003, *Mycetomoellerius urichii* (Forel, 1893), *Camponotus zonatus* Emery, 1894, and *Pogonomyrmex mayri* Forel, 1899, whose relative abundance ranged from 40 and 60%. Litter-associated ants showed a less dominant structure, in addition to greater variability with respect to the most frequent species between sites (Figure 3b). In the case of the N01 site, the species *Solenopsis bicolor* Emery, 1896, *Anochetus inermis* André, 1889, *Paratrachymyrmex irmgardae* (Forel, 1912), and *Pheidole guajirana* correspond to the most frequent species (60–80%); in the N02 and N03 sites, the most abundant species exhibited a moderate capture frequency (50–60%), while in the N04 site, *E. ruidum* was dominant (80%).

*3.3. Spatial and Temporal Variation*

The variations in the composition and abundance of the ants that forage on the ground are associated with the sites (R = 1.0; $p < 0.01$) and not with the climatic seasons (R = 0.20; $p = 0.27$) (Figure 4a). Litter ants show a more heterogeneous trend, although the variation seems to be more associated with climatic seasons (Figure 4b); however, there are differences in composition and abundance between the four sites (R = 0.47; $p = 0.01$) but not between seasons (R = 0.28; $p = 0.16$).

Analysis of beta diversity (βjac) revealed a differentiation between moderate to high (50–70%) for the set of fragments evaluated and mainly for the ants of the arboreal vegetation and leaf litter (Figure 5). Among sites, the values oscillated between 40 and 80% (Table 1). This variation corresponds mainly to turnover processes, a trend that is maintained for the three microhabitats (Table 1). The variation explained by richness differences

(gain or loss of species) was in most cases low for the pairs of sites, although in the specific case of N04, this component of beta diversity tended to represent around 50% of the differentiation (Table 1).

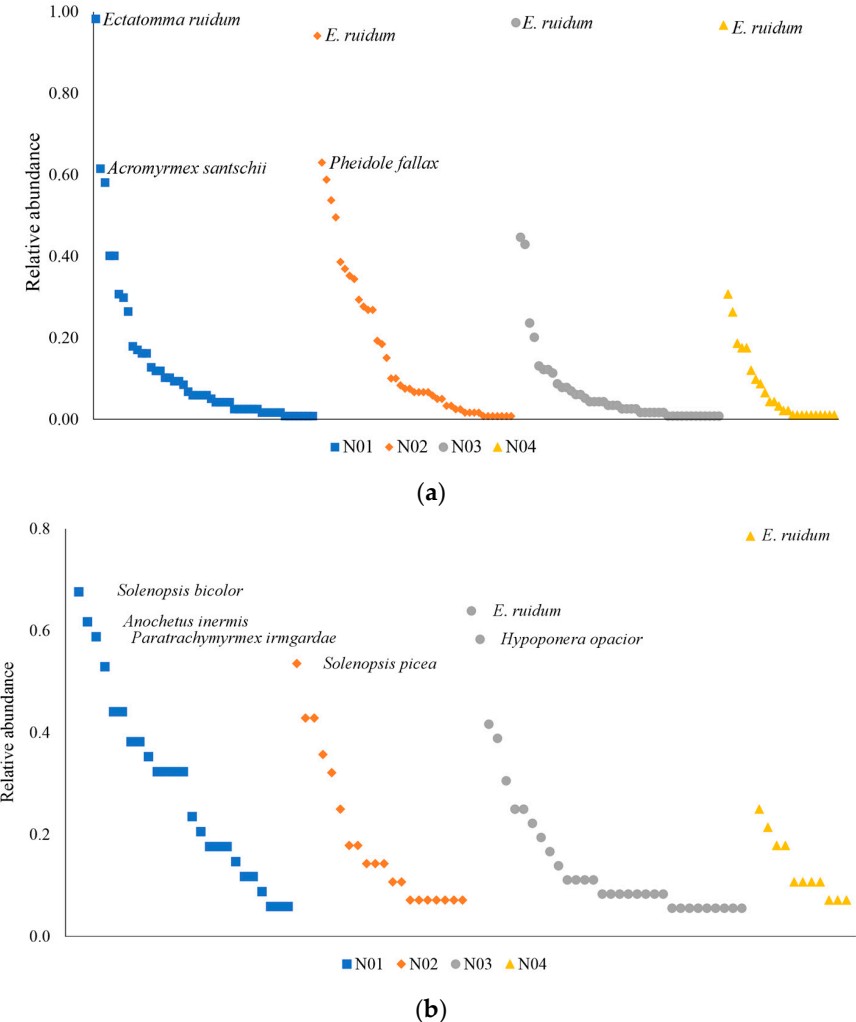

**Figure 3.** Rank–abundance curves showing the distribution of capture frequencies (relative abundance) for the ant assemblage in four urban fragments of the TDF in the city of Santa Marta. (**a**) Soil ants ("pitfall"); (**b**) ants associated with leaf litter (Winkler). Species with capture frequencies ≥ 0.50 are indicated.

**Table 1.** Beta diversity and its components of turnover (below the diagonal) and richness differences (above the diagonal) among sites. Values close to 1 correspond to the greatest turnover or richness differences between each of the sites. The values on the diagonal correspond to the total number of species recorded for each locality.

| Beta Diversity | N01 | N02 | N03 | N04 |
|---|---|---|---|---|
| Ground (pitfall) | | | | |
| N01 | 50 | 0.10 | 0.08 | 0.44 |
| N02 | 0.36 | 44 | 0.02 | 0.36 |
| N03 | 0.36 | 0.45 | 45 | 0.40 |
| N04 | 0.15 | 0.24 | 0.09 | 26 |
| Leaf litter (Winkler) | | | | |
| N01 | 38 | 0.06 | 0.06 | 0.52 |
| N02 | 0.48 | 35 | 0.12 | 0.51 |
| N03 | 0.45 | 0.36 | 41 | 0.39 |
| N04 | 0.27 | 0.21 | 0.14 | 15 |

**Table 1.** *Cont.*

| Beta Diversity | N01 | N02 | N03 | N04 |
|---|---|---|---|---|
| Vegetation (hand collection) | | | | |
| N01 | 26 | 0.17 | 0.08 | 0.50 |
| N02 | 0.51 | 20 | 0.09 | 0.32 |
| N03 | 0.63 | 0.65 | 23 | 0.30 |
| N04 | 0.14 | 0.40 | 0.30 | 12 |

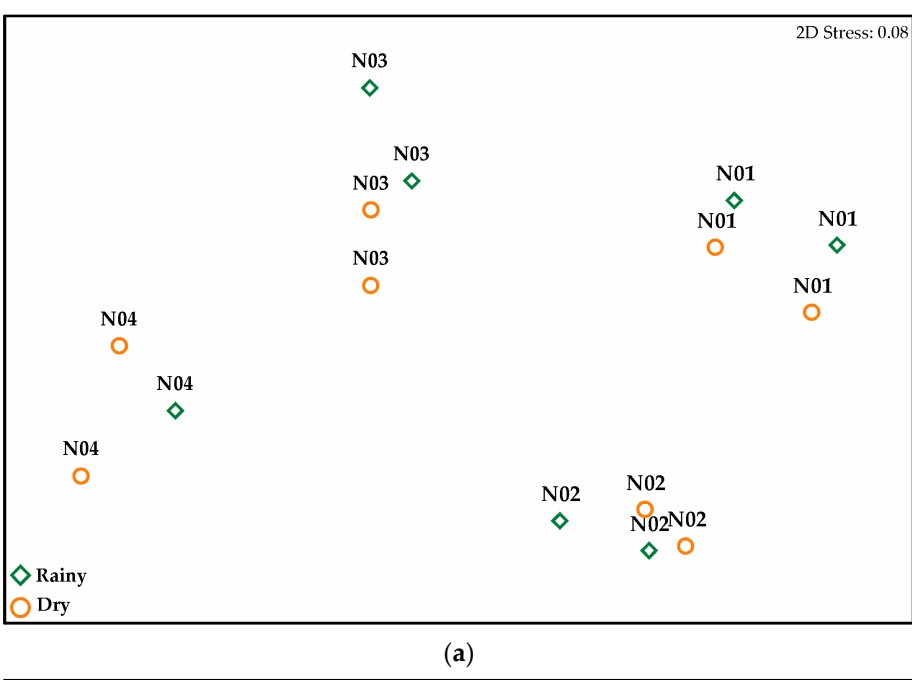

(**a**)

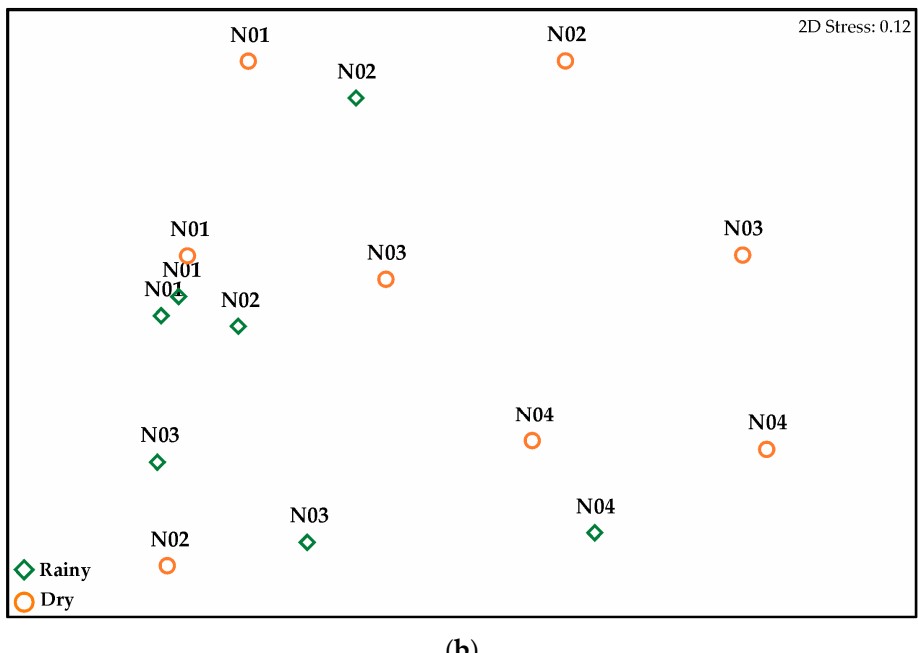

(**b**)

**Figure 4.** Ordination analysis using non-metric multidimensional scaling (nMDS) for the assembly of ants: (**a**) that forage on the ground and (**b**) are associated with leaf litter. There is a greater variation associated with the sampling locations and less differentiation between the rainy (diamonds) and dry (circles) seasons.

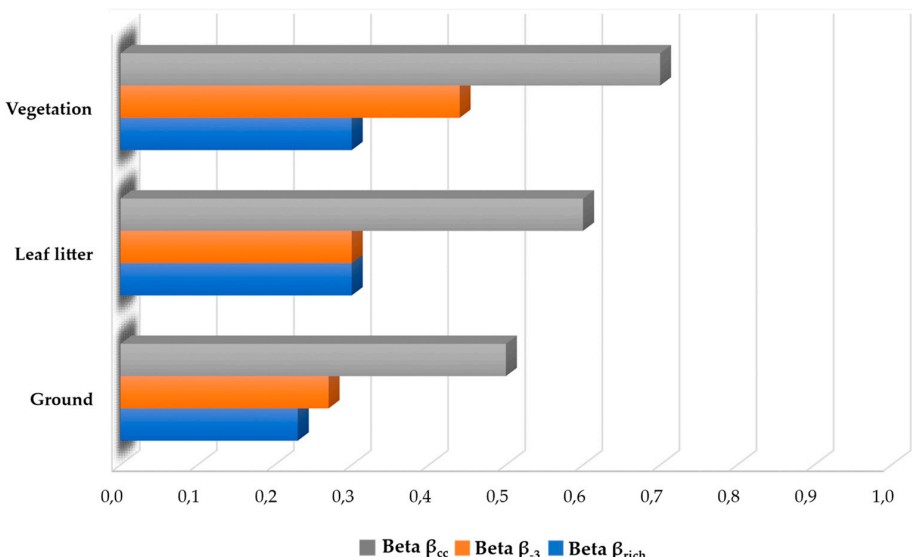

**Figure 5.** Beta diversity ($\beta_{cc}$) for the assemblage of soil, vegetation, and litter-associated ants, based on the Jaccard index (Jac) and its turnover ($\beta_{-3}$) and richness differences ($\beta_{rich}$) components.

### 3.4. Functional Groups

Seventeen functional groups were identified, homogeneously distributed among the sites, except for N04 with only ten groups (Figure 6). In the latter, specialist groups were less conspicuous, with the specialist groups being the least conspicuous (Appendix A). The predominant functional groups corresponded to the omnivores and predators of large and small sizes, both arboreal and on the ground, followed by dacetine predator ants and higher-agriculture ants (Appendix A). There was no variation for the functional groups between the rainy and dry seasons.

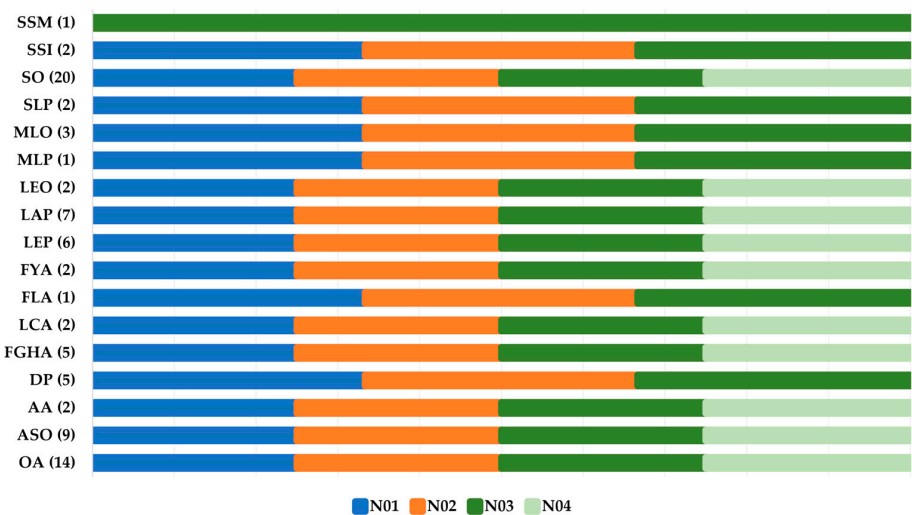

**Figure 6.** Spatial variation in the functional groups recorded in the urban fragments of TDF. The number of species associated with each functional group category is shown in parentheses. The functional groups correspond to the following: fungus leaf-cutter agriculture (LCA); fungus lower-agriculture (FLA); fungus yeast agriculture (FYA); fungus-generalized higher-agriculture (FGHA); dacetine predator (DP); large/medium-sized arboreous generalist predator (LAP); large epigeic generalist predator (LEP); medium-sized litter/hypogeic generalist predator (MLP); small epigeic/litter/hypogeic generalist predator (SLP); soil-specialized isopoda forager/predator (SSI); soil-specialized millipede predator (SSM); army ants (AA), arboreous omnivore (AO); small epigeic/litter/hypogeic omnivore (SO); arboreous/soil omnivore (ASO); large/medium-sized epigeic omnivore (LEO); medium-sized litter/hypogeic omnivore (MLO).

## 4. Discussion

### 4.1. Composition and Completeness of Sampling

The ant fauna collected in the urban fragments of the TDF in the city of Santa Marta represent around 64% of the subfamilies registered for the Neotropics and Colombia, as well as 35% of the genera in Colombia [40]. Of the 84 determined species, 77 were identified at the species level, corresponding to 7% of the current records for the country [40,41].

The contribution of the Myrmicinae subfamily to the ant assemblage in the TDF in Colombia (50% of the species) is a relatively constant trend [13,16,42,43], a pattern explained by the high taxonomic richness and biological diversity [44]. In this study, the high richness in Myrmicinae is mainly contributed by genera that use diverse habitat resources, such as the arboreal layer (*Cephalotes* and *Crematogaster*); the epigeal layer (*Pheidole* and *Solenopsis*); and the litter (*Strumigenys*). Other subfamilies in order of importance correspond to Ponerinae, Formicinae, Dolichoderinae, Dorylinae, and Ectatomminae; however, the contribution to richness varies among the different studies.

One of the main limitations of the available ant composition data from the TDF is the level of taxonomic identification, making it difficult to compare information at different geographic scales (e.g., beta diversity analysis). In this study, more than 90% of the species were identified at a specific level, allowing the recognition of four new records of ants for the country: *Hypoponera clavatula* (Emery, 1906); *Myrmicocrypta buenzlii* Borgmeier, 1934; *Solenopsis whitfordi* Mackay et al., 2013; and *Strumigenys tanymastax* (Brown, 1964). The results suggest that the urban fragments of the TDF in Santa Marta are acting as a reservoir of ant diversity, the study of which yields an increase in the knowledge of the country's ants and the biogeographical and evolutionary understanding of some genera [19].

Sampling coverage values interpreted in terms of completeness suggest, in the case of pitfall traps and baits (>93%), a representative sample for the assembly of ants that forage on the ground in the TDF fragments studied. In the case of ants associated with leaf litter, the sampling deficit values are higher, indicating a greater probability that by increasing the sample size, a new sampled individual corresponds to a different species [31]. In the case of litter, it is important to consider aspects such as the quality and quantity of litter available in the TDF, which tend to decrease with drought; furthermore, leaf litter ants tend to be less numerous with a higher number of unique species, biasing the results of the sampling coverage estimator [30].

### 4.2. Alpha Diversity

Ants have been widely studied in the Colombian Caribbean region [16,42,43,45]; however, there is little information available on the ecological patterns of this group in the TDF. In this study, the diversity expressed as the effective number of species shows a marked spatial and not temporal variation (i.e., climatic seasons). This trend is consistent for all microhabitats considered, in which the diversity exhibits higher values of common species ($^1D$) and dominant species ($^2D$) for the TDF fragments located within the urban area of the city (N01 and N02), with respect to those of the peri-urban zone (N03 and N04); however, the variations at the level of richness ($^0D$) are not as marked between the four sites, except N04 which is the least diverse (Figure 2).

Ant diversity assessments in some TDF fragments of the Colombian Caribbean show different trends in relation to diversity in space and time, contrasting notably with the results of this study. In the northern zone of La Guajira, there is a high spatial variability between the characterized fragments and no difference in richness between the seasons [43]; meanwhile, in the Atlántico department, the results suggest a greater variability in diversity attributable to climatic seasons [16]. However, for the hills that surround the city of Santa Marta, there were no spatial or temporal variations in the richness and diversity of ants [45]. These differences may be strongly associated with the ecological and environmental dynamics of the evaluated sites (e.g., anthropic pressures), which are poorly understood for most biological groups within the TDF ecosystem [21], including ants.

In this study, the structure of the ant assemblages presents a distribution of their capture frequencies (relative abundance) like a logarithmic series curve. Within the group of dominant species is *Ectatomma ruidum*, which has been related to highly disturbed environments in TDF landscapes of Valle del Cauca [46] and considered a potential indicator of low species richness due to their high dominance [17]. Our results suggest that *E. ruidum* dominance does not consistently decrease ant species richness since the number of species that forage on the ground was similar between sites N01, N02, and N03. Although N04 registered a substantial reduction in the richness of soil ant species, this decrease could be due to factors associated with habitat quality and scarcity of microhabitats, making it difficult for other species to settle. *E. ruidum* was also dominant (<80%) between litter ants in N03 and N04 and subdominant (<50%) in N01 and N02. In general, the dominance of *E. ruidum* could be shaped mainly by the vegetation cover (habitat) and the temperature (climate) in the TDF fragments studied [47] and not by the ability to displace other species to appropriate resources in the environment.

Another relatively dominant species between ground ants in the TDF was *Pheidole fallax*, except in N01. The dominance of *P. fallax* in most of the fragments could be explained by the nesting and foraging preferences of this species, which develop strictly on the ground in environments with open canopies and high temperatures, where *P. fallax* is more aggressive and has better ecological performance [48]. These characteristics are found in fragments N02, N03, and N04, where *P. fallax* was more frequent in the open areas of the fragments but relatively rare inside them.

### 4.3. Spatial and Temporal Variation

In general terms, the results of other research do not yield a clear pattern at the spatial or temporal level [16,45], although the balance seems to lean towards a trend in high spatial differentiation [42,43], as suggested by the results in our study. In this sense, high levels of beta diversity seem to be a distinctive feature in the TDF, possibly related to environmental selection pressures because of marked seasonality [1]. In the case of the ants of the urban fragments studied, the number of exclusive species corresponds to 30% of the fauna recorded for the set of localities, revealing the value of importance that research represents at local scales and even in isolated fragments. This is a key tool for the integral management and conservation of the TDF in Colombia since, at present, this ecosystem is distributed in very small fragments in the country [2].

The low diversity registered in the locality of Quebrada Seca (N04) does not necessarily reflect a deficient state of conservation since the richness of species described for this site is comparable to that registered by [46] in two hills of the city (La Llorona and La Cueva); there, the arid conditions caused by low general rainfall (8–10 months of drought) and geomorphological characteristics of the soil seem to limit the establishment of a greater variety of ants. According to this, the ants living in the hills that surround the city, including the Quebrada Seca sector (N04), are adequate to the environmental and habitat conditions offered. However, the effect of the disturbance in this locality cannot be completely ruled out due to the human settlements in its surroundings and the extractive activities in the forest for the livelihood of these communities. In this sense, one of the main problems and challenges that the city faces in urban aspects is the unplanned and disproportionate growth in the hills that surround it, becoming a latent threat to the biodiversity of the TDF.

### 4.4. Functional Groups

The set of functional groups in the urban fragments of the TDF in the city of Santa Marta reflect a great variability in resources available for the establishment of numerous ant species, trophic requirements (herbivores, omnivores, predators), and microhabitats (leaf litter, vegetation, ground) differentials. In addition, the homogeneous distribution of the functional groups between the TDF fragments studied suggests that although there is a differentiation in the taxonomic composition of the ant fauna, the function of the species in a particular fragment may be carried out by ecologically equivalent species in

another fragment. Likewise, the presence of 12 ecologically demanding functional groups suggests that the TDF fragments provide the resources for the establishment of species with specialized behaviors and resource specificity, such as army ants (AA) or specialized predatory ants (SSM), such as the *Thaumatomyrmex* sp. *mutilatus* group.

The opposite behavior between the beta diversity and the functional groups for the ants that inhabit the urban fragments of the TDF in Santa Marta highlights the importance of considering the joint analysis of the taxonomic and functional diversity of the ants, since each component reflects different facets within the ecosystem. Unfortunately, conservation efforts are often oriented based on species richness, promoting priority areas with a greater number of species (excluding other community or focus group properties); however, other components of biodiversity, such as functional diversity, together with taxonomic valuation, yield better approximation to the ecological dynamics of highly heterogeneous ecosystems (i.e., environmental, and biological dynamics), such as the TDF.

Faced with a scenario of loss of biodiversity, such as the one currently being experienced, the decrease in species richness may result in a decrease in the levels of functioning of the ecosystem; however, this effect is closely related to changes in community composition and the ecosystem functions involved, resulting in the loss of keystone species. The integration of different dimensions of biodiversity, including those that involve ecological (functional diversity) and historical aspects of the biota (phylogenetic diversity), can provide crucial information for the application and success of conservation strategies.

## 5. Conclusions

The taxonomic diversity of ants in the urban fragments of the TDF in Santa Marta (Colombia) show a marked spatial variation, without being influenced by the rainy and dry seasons. The latter is reflected at the level of species richness both in the set of ants that live in the litter, and in those that nest and forage on the ground. The results also suggest that the humidity in the environment can be a limiting factor for the ants of the TDF since the number of individuals in each of these groups is influenced by the presence of rain. Based on the above, the abundance of ants that live in the litter drops drastically by half during the time of water deficit, while those that forage on the ground increase by approximately two orders of magnitude. Likewise, there are high levels of species turnover between the urban fragments of the TDF, which translates into relatively different assemblages between them; despite taxonomic turnover, there is a broad similarity in functional groups between the four fragments, indicating that differences in taxonomic composition between them occur from the replacement of functionally similar species (i.e., ecological equivalence of species in each ant assemblage).

The tropical dry forest fragments evaluated in the city of Santa Marta are areas that promote and maintain the diversity of ants. Cases such as the TDF plot of the University of Magdalena (N01), Quinta de San Pedro Alejandrino (N02), and La Iguana Verde (N03) reveal that betting on the regeneration and conservation of these sites, under practices such as ecological restoration, sustainable ecotourism, and environmental pedagogy, offer results that allow the interaction of the urban environment (even with its adverse effects) with natural areas, such as the highly sensitive tropical dry forest.

**Author Contributions:** L.M.R.O. and R.J.G. contributed equally to the conceptualization and development of this research. The sampling design and field work were carried out by L.M.R.O. and R.J.G. The taxonomic identification of the ants was carried out by R.J.G., while the validation, data curation, and analysis of the information were carried out by L.M.R.O. L.M.R.O. prepared the first draft of the manuscript. L.M.R.O. and R.J.G. reviewed and edited the final version of the manuscript. R.J.G. oversaw obtaining funds for this research. All authors have read and agreed to the published version of the manuscript.

**Funding:** This research was funded by Ministerio de Ciencia, Tecnología en Innovación de Colombia (Minciencias), ICETEX, and Universidad del Magdalena, grant number 2021–1029. The APC was funded by Vicerrectoría de Investigación, Universidad del Magdalena.

**Institutional Review Board Statement:** Not applicable.

**Data Availability Statement:** The data presented in this study are openly available in [repository name e.g., SIB Colombia] at [https://doi.org/10.15472/2cdkza] (accessed on 2 August 2021).

**Acknowledgments:** Thanks to the directives of the Quinta San Pedro Alejandrino and the TDF plot of Unimagdalena for allowing access. To Luz Adriana Velazco from the Iguana Verde Natural Reserve and Libardo López in Quebrada Seca for access to the TDF within their properties. The Centro de Colecciones Científicas of the Universidad del Magdalena for logistical support. Also, to Hubert Sierra, Johan Roncallo and José Juan Camargo for their support in the field work. Many thanks to three anonymous reviewers for their comments and suggestions that greatly improved this manuscript.

**Conflicts of Interest:** The authors declare no conflict of interest. The funders had no role in the design of the study; in the collection, analyses, or interpretation of data; in the writing of the manuscript; or in the decision to publish the results.

## Appendix A

**Table A1.** List of ant species collected in four urban fragments of the TDF in the city of Santa Marta. The method or methods by which each species was collected is indicated, as well as the functional group to which it belongs and the frequency of capture (%) calculated as the appearance of a species with respect to the total number of sampling units used in the study (see Materials and Methods section). Baits (Ba), hand collection (Hc), mini-Winkler bags (mW), and pitfall traps (Pf). The functional groups correspond to the following: fungus leaf-cutter agriculture (LCA); fungus lower-agriculture (FLA); fungus yeast agriculture (FYA); fungus-generalized higher-agriculture (FGHA); dacetine predator (DP); large/medium-sized arboreous generalist predator (LAP); large epigeic generalist predator (LEP); medium-sized litter/hypogeic generalist predator (MLP); small epigeic/litter/hypogeic generalist predator (SLP); soil-specialized isopoda forager/predator (SSI); soil-specialized millipede predator (SSM); army ants (AA); arboreous omnivore (AO); small epigeic/litter/hypogeic omnivore (SO); arboreous/soil omnivore (ASO); large/medium-sized epigeic omnivore (LEO); medium-sized litter/hypogeic omnivore (MLO).

| Subfamily | Species | Functional Group | N01 | N02 | N03 | N04 | Relative Abundance (%) |
|---|---|---|---|---|---|---|---|
| Myrmicinae | *Acromyrmex octospinosus* (Reich, 1793) | LCA | | | Hc, Pf | | 0.7 |
| | *Acromyrmex santschii* (Forel, 1912) | LCA | Ba, Hc, mW, Pf | Hc, Pf | Hc, mW, Pf | Pf | 19.2 |
| | *Cephalotes* aff. *bimaculatus* | AO | | Hc | | | 0.1 |
| | *Cephalotes femoralis* (Smith, 1853) | AO | Hc, Pf | Hc, Pf | Hc, mW, Pf | Hc, Pf | 1.7 |
| | *Cephalotes minutus* (Fabricius, 1804) | AO | Ba, Hc, mW, Pf | Hc, Pf | mW, Pf | Hc, Pf | 3.9 |
| | *Cephalotes pellans* De Andrade, 1999 | AO | Hc, mW, Pf | | mW, Pf | | 0.7 |
| | *Cephalotes pusillus* (Klug, 1824) | AO | Hc, mW, Pf | Hc, mW, Pf | Hc, mW, Pf | Hc, mW, Pf | 8.1 |
| | *Cephalotes* sp. | AO | mW | | | | 0.1 |
| | *Crematogaster crinosa* Mayr, 1862 | AO | Hc, Pf | Ba, Hc, mW, Pf | Hc, Pf | Ba, Hc, Pf | 4.2 |
| | *Crematogaster distans* Mayr, 1870 | AO | Hc, mW | | | | 0.2 |
| | *Crematogaster limata* Smith, 1858 | AO | | | Hc | | 0.1 |
| | *Crematogaster obscurata* Emery, 1895 | OA | Hc, mW, Pf | | Hc, mW, Pf | mW, Pf | 3.8 |
| | *Crematogaster rochai* Forel, 1903 | AO | | | Hc, Pf | | 0.4 |
| | *Crematogaster torosa* Mayr, 1870 | AO | Hc | | Hc, Pf | Ba, Hc, Pf | 2.2 |
| | *Cyphomyrmex flavidus* Pergande, 1896 | FYA | | mW, Pf | | | 0.4 |
| | *Cyphomyrmex rimosus* (Spinola, 1851) | FYA | Hc, mW, Pf | Pf | mW, Pf | Pf | 5.9 |
| | *Megalomyrmex silvestrii* Wheeler, 1909 | MLO | mW | | | | 0.1 |
| | *Mycetomoellerius urichii* (Forel, 1893) | FGHA | Ba, Hc, mW, Pf | mW, Pf | Hc, mW, Pf | Pf | 11.1 |
| | *Mycetomoellerius zeteki* (Weber, 1940) | FGHA | | | Hc, mW, Pf | | 0.3 |
| | *Myrmicocrypta buenzlii* Borgmeier, 1934 | CHI | mW, Pf | mW, Pf | Pf | | 4.7 |
| | *Nesomyrmex* sp. n. | AO | Pf | | | | 0.1 |
| | *Paratrachymyrmex cornetzi* (Forel, 1912) | FGHA | mW, Pf | mW, Pf | Pf | | 4.1 |
| | *Paratrachymyrmex irmgardae* (Forel, 1912) | FGHA | Ba, Hc, mW, Pf | mW | mW | | 3.1 |
| | *Pheidole distorta* Forel, 1899 | SO | mW, Pf | Ba, Pf | Pf | | 1.8 |
| | *Pheidole fallax* Mayr, 1870 | SO | Ba, Hc, mW, Pf | Ba, Pf | Ba, mW, Pf | Ba, Hc, Pf | 24.0 |
| | *Pheidole guajirana* Wilson, 2003 | SO | Ba, Hc, mW, Pf | Ba, mW, Pf | Ba, mW, Pf | mW, Pf | 16.2 |

**Table A1.** *Cont.*

| Subfamily | Species | Functional Group | N01 | N02 | N03 | N04 | Relative Abundance (%) |
|---|---|---|---|---|---|---|---|
| | *Pheidole impressa* Mayr, 1870 | SO | | Pf | | | 0.2 |
| | *Pheidole inversa* Forel, 1901 | SO | Ba, mW, Pf | Ba, mW, Pf | Ba, mW, Pf | Ba, Hc, Pf | 9.4 |
| | *Pheidole leptina* Wilson, 2003 | SO | | Ba, Hc | | | 0.2 |
| | *Pheidole praeusta* Roger, 1863 | SO | Ba | | | | 0.1 |
| | *Pheidole subarmata* Mayr, 1884 | SO | | Pf | | Ba, Pf | 1.5 |
| | *Pheidole urbana* Camargo y Guerrero, 2020 | SO | Pf | Ba, Pf | Ba, Pf | mW | 7.3 |
| | *Pheidole* sp. 10. | SO | Hc, mW, Pf | | | | 0.4 |
| | *Pogonomyrmex mayri* Forel, 1899 | LEO | Ba, Pf | Ba, mW, Pf | Ba, mW, Pf | Ba, mW, Pf | 17.4 |
| | *Rogeria curvipubens* Emery, 1894 | SO | | mW | | | 0.1 |
| | *Rogeria foreli* Emery, 1894 | SO | mW, Pf | mW | mW | | 2.2 |
| | *Sericomyrmex bondari* Borgmeier, 1937 | FGHA | | | mW, Pf | | 0.7 |
| | *Solenopsis altinodis* Forel, 1912 | SO | Ba, Hc, mW, Pf | Ba, Hc, mW, Pf | mW | | 8.4 |
| | *Solenopsis bicolor* (Emery, 1906) | SO | Ba, Hc, mW, Pf | mW, Pf | mW, Pf | mW | 10.0 |
| | *Solenopsis geminata* (Fabricius, 1804) | LEO | Ba | mW, Pf | Hc, Pf | mW | 1.4 |
| | *Solenopsis picea* Emery, 1896 | SO | mW, Pf | mW, Pf | mW, Pf | mW, Pf | 7.4 |
| | *Solenopsis whitfordi* Mackay et al., 2013 | SO | mW | mW | mW, Pf | mW, Pf | 1.3 |
| | *Strumigenys dyseides* Bolton, 2000 | DP | Pf | | | | 0.1 |
| | *Strumigenys eggersi* Emery, 1890 | DP | | mW | mW | | 0.2 |
| | *Strumigenys elongata* Roger, 1863 | DP | mW | mW | mW | | 1.3 |
| | *Strumigenys spathula* Lattke & Goitía, 1997 | DP | Pf | | mW | | 0.5 |
| | *Strumigenys tanymastax* (Brown, 1964) | DP | mW | mW | mW | | 1.7 |
| | *Temnothorax subditivus* (Wheeler, 1903) | SO | Ba, Hc, mW, Pf | Ba, mW, Pf | mW, Pf | mW, Pf | 9.2 |
| | *Trichomyrmex destructor* (Jerdon, 1851) | ASO | | Hc, Pf | | | 0.3 |
| | *Wasmannia auropunctata* (Roger, 1863) | SO | Ba, Pf | | Ba, mW, Pf | | 1.5 |
| Ponerinae | *Anochetus inermis* André, 1889 | MLP | mW, Pf | Ba, mW, Pf | mW | | 7.6 |
| | *Hypoponera clavatula* (Emery, 1906) | SLP | mW | | | | 0.6 |
| | *Hypoponera opacior* (Forel, 1893) | SLP | mW, Pf | mW | mW | | 3.8 |
| | *Leptogenys pubiceps* Emery, 1890 | SSI | Hc, Pf | mW, Pf | Pf | | 2.2 |
| | *Leptogenys ritae* | SSI | mW, Pf | | | | 0.2 |
| | *Odontomachus bauri* Emery, 1892 | LEP | Ba, mW, Pf | Ba, mW, Pf | Hc, mW, Pf | Pf | 7.9 |
| | *Odontomachus ruginodis* Wheeler, 1908 | LEP | mW, Pf | Hc, mW, Pf | mW, Pf | | 3.5 |
| | *Pachycondyla harpax* (Fabricius, 1804) | LEP | Ba, mW, Pf | mW | mW, Pf | Pf | 2.5 |
| | *Platythyrea pilosula* (Smith, 1858) | LEP | | | Pf | Pf | 0.3 |
| | *Thaumatomyrmex* sp. *mutilatus* group | SSM | | | Hc | | 0.1 |
| Formicinae | *Brachymyrmex cordemoyi* Forel, 1895 | ASO | mW | | | | 0.2 |
| | *Brachymyrmex minutus* Forel, 1893 | ASO | Ba, Hc, mW, Pf | Hc, mW, Pf | mW, Pf | Ba, mW, Pf | 6.6 |
| | *Camponotus blandus pronotalis* Santschi, 1936 | ASO | Ba, Pf | Ba, Hc, mW, Pf | mW | Ba | 2.1 |
| | *Camponotus zonatus* Emery, 1894 | ASO | Pf | Hc, Pf | mW, Pf | Pf | 12.5 |
| | *Camponotus coruscus* (Smith, 1862) | ASO | Pf | | Ba, mW, Pf | | 1.5 |
| | *Camponotus lindigi* Mayr, 1870 | ASO | Ba, Hc, Pf | Ba, Hc, mW, Pf | Ba, Hc, mW, Pf | Cm, mW, Pf | 11.6 |
| | *Camponotus* sp. 5 | ASO | | | Pf | | 0.1 |
| | *Nylanderia nodifera* (Mayr, 1870) | MLO | | mW, Pf | Ba, mW, Pf | | 0.7 |
| | *Paratrechina longicornis* (Latreille, 1802) | MLO | Pf | Ba, Cm, Pf | Ba, Hc, Pf | | 2.0 |
| Pseudomyrmecinae | *Pseudomyrmex boopis* (Roger, 1863) | LAP | mW, Pf | mW, Pf | Ba, Hc, mW, Pf | Ba, Pf | 3.8 |
| | *Pseudomyrmex elongatus* (Mayr, 1870) | LAP | Hc, Pf | | | Hc | 0.8 |
| | *Pseudomyrmex gracilis* (Fabricius, 1804) | LAP | | | Hc | | 0.1 |
| | *Pseudomyrmex simplex* (Smith, 1877) | LAP | Hc, Pf | Hc, Pf | Hc | | 1.0 |
| | *Pseudomyrmex urbanus* (Smith, 1877) | LAP | Pf | Hc, mW, Pf | Hc | Hc | 0.8 |
| | *Pseudomyrmex venustus* (Smith, 1858) | LAP | Hc | Pf | Hc, Pf | Hc | 0.7 |
| Dolichoderinae | *Dolichoderus diversus* Emery, 1894 | AO | | mW | | | 0.1 |
| | *Dorymyrmex biconis* Forel, 1912 | SO | mW | Hc, Pf | | | 0.9 |
| | *Forelius damiani* Guerrero y Fernández, 2008 | SO | | Ba, Pf | mW, Pf | | 1.6 |
| | *Tapinoma melanocephalum* (Fabricius, 1793) | ASO | Ba, Hc, Pf | Hc | | | 1.2 |
| Ectatomminae | *Ectatomma* aff. *ruidum* | LEP | | Pf | | | 0.1 |
| | *Ectatomma ruidum* (Roger, 1860) | LEP | Ba, Hc, mW, Pf | Ba, Hc, mW, Pf | Ba, Hc, mW, Pf | Ba, Hc, mW, Pf | 81.2 |
| | *Ectatomma tuberculatum* (Olivier, 1792) | LAP | | | Hc | | 0.1 |
| Dorylinae | *Labidus coecus* (Latreille, 1802) | AA | Pf | | mW, Pf | mW, Pf | 2.0 |
| | *Neivamyrmex iridescens* Borgmeier, 1950 | AA | | Pf | | | 0.2 |

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
