# Peer review of "Spatial Turnover and Functional Redundancy in the Ants of Urban Fragments of Tropical Dry Forest"

_diversity, doi:10.3390/d15070880_

Round 1

Reviewer 1 Report

While I haven't been able to conduct a thorough review due to time constraints, I was able to skim through the MS and read sections that concern my expertise as a myrmecologist.  Regarding ant sampling: I was a little surprised to read that pitfall traps were filled in part with ethanol. In my experience, in pitfall traps a liquid is used that doesn't deter ants, commonly that is soapy water or antifreeze. I wonder on what grounds this decision was made and I worry that through that some ground-dwelling species could have been missed (which is simply a limitation). 

The English requires some minor edits, MS should be reviewed by a native speaker or equivalent. Some words, expressions and phrases should certainly be revised.

e.g. "capture" should be replaced with "collection" in most places; "leaving ... to act" is not a proper expression in this context, ...

Author Response

Reviewer 1 (reviewer comments indicated in italics)

  • I was a little surprised to read that pitfall traps were filled in part with ethanol. In my experience, in pitfall traps a liquid is used that doesn't deter ants, commonly that is soapy water or antifreeze. I wonder on what grounds this decision was made and I worry that through that some ground-dwelling species could have been missed (which is simply a limitation).

We use pitfall traps filled with pure ethanol (96%) for several reasons: to increase the preservation of the specimens, since the sites where the study was carried out present high temperatures that easily lead to tissue damage. The optimal preservation of the specimens, while they are in the pitfall traps, allows us to preserve the DNA for future studies that we are carrying out.

Although ethanol can be a deterrent for some animals, field experience has allowed us to see that ants do not avoid pitfall traps, being collected in abundance. The same could be true of the containers used at the bottom end of winkler bags; although these containers are filled with Ethanol, the ants are always extracted.

  • The English requires some minor edits, MS should be reviewed by a native speaker or equivalent. Some words, expressions and phrases should certainly be revised.

e.g. "capture" should be replaced with "collection" in most places; "leaving ... to act" is not a proper expression in this context.

The word "collection" was adjusted in two of the cases: line 108 and 175. In the other parts of the text, the use of the word "capture" was maintained since its use is quite common in the context of express the frequency of capture of ants.

The text of the expression "leaving ... to act" was also revised and adjusted: lines 117-118.

Other minor spelling changes were reviewed and adjusted.

Reviewer 2 Report

The study presented in the manuscript addresses an important research question and provides significant insights into urban fragments of tropical dry forest. The authors have conducted a comprehensive and well-designed investigation, employing appropriate methodologies and data analysis techniques. The experimental design is robust, and the methods are clearly described, allowing for reproducibility.

I have only two Minor Concern, one related with the Literature Review and other related with the Beta Diversity Partition used.

 1. Provide other example studies for this kind of habitat in addition to Caribbean and Colombia.

2. The Carvalho approach, as described in the article by Carvalho et al. (2013), partitions beta diversity into two components: turnover (due to species replacement) and richness differences (due to differences in species richness between sites). This approach provides a more ecologically meaningful way of understanding beta diversity than the Baselga approach that gives emphasis to nestedness.

Nestedness is a component of beta diversity that measures the degree to which species assemblages in smaller, more isolated habitats represent proper subsets of those found in larger, more connected habitats. In other words, nestedness reflects the extent to which species richness declines in smaller, isolated habitats due to the absence of some species that are present in larger, more connected habitats. Nestedness is important for understanding how habitat fragmentation and loss can affect biodiversity patterns, and for identifying sites that are particularly vulnerable to species loss.

On the other hand, richness differences reflect the absolute difference in species richness between two sites. Richness differences are important for understanding how the total number of species in a community changes across different habitats or geographic regions, but do not necessarily reflect differences in the species composition or structure of those communities.

The Beta partition approach proposed by Carvalho et al. (2011, 2013) is considered more adequate than the Baselga approach due to several reasons: Most crucial is scaling, because the species replacement fractions ( β sim and β jtu )  are not standardized the same way as the overall dissimilarity measures ( β sor and β cc , respectively) from which they are subtracted. In fact, the choice of the maximum possible values for such scaling is not substantiated because there are several other, equally if not more acceptable possibilities.

So, in the context of beta diversity partitioning, using nestedness as a component measures the degree to which small, isolated habitats differ from larger, connected habitats in terms of their species composition, while using richness differences as a component measures the overall difference in species richness between two sites, regardless of their compositional similarity or differences.

In summary, the Beta partition approach of Carvalho et al. is considered more adequate than the Baselga approach due to its flexibility, accommodation of non-linear relationships, improved discrimination, increased predictive power, and biologically meaningful interpretation. Both nestedness and richness differences can be useful components of beta diversity partitioning, but they provide different information about the underlying ecological processes driving beta diversity

To measure nestedness, the authors can use a dedicated method like the NDOF.

Carvalho, J.C., Cardoso, P., Borges, P.A.V., Schmera, D. & Podani, J. (2013). Measuring fractions of beta diversity and their relationships to nestedness: a theoretical and empirical comparison of novel approaches. Oikos, 122: 825–834. DOI:10.1111/j.1600-0706.2012.20980.x

Some improvements are needed in the Introduction

Author Response

Reviewer 2 (reviewer comments indicated in italics)

Provide other example studies for this kind of habitat in addition to Caribbean and Colombia.

  • The requested adjustment was made.

The Carvalho approach, as described in the article by Carvalho et al. (2013), partitions beta diversity into two components: turnover (due to species replacement) and richness differences (due to differences in species richness between sites). This approach provides a more ecologically meaningful way of understanding beta diversity than the Baselga approach that gives emphasis to nestedness.

Reviewer 2's suggestions were very important, allowing us to improve data analysis. We reviewed the suggested literature and understood the methodological aspects commented by the reviewer, so we opted to make changes based on Carvalho's proposal.

  • Changed beta diversity analysis citations: lines 146-150; 527-530.
  • Figure 5 was changed, according to the changes regarding Carvalho's beta diversity metric.

The content of Table 1 with the updated values using the Carvalho metric was modified.

Reviewer 3 Report

The manuscript deals with the community analysis of the ant fauna in two sites within Santa Marta (Colombia) and in two close sites outside the urban area. The area is subjected to hard urbanization and to a potentially steep increase in fragmentation. The authors collected ants by different sampling methods and analyzed the data by assessing the diversity and species assemblage, also from a functional group point of view. 

I found the manuscript well-written, the study well-conducted, and the analyses correct and exhaustive. Searching in the literature, the paper is indeed one of the few dealing with the ant community in this area, and it deserves to be published. I have only a few minor comments that are reported as notes in the attached file.

Author Response

Reviewer #3

  • Reviewer 3's comments are in the PDF document he attached. We expose below the responses to his comments and suggestions.
  • The acronym in English for tropical dry forest was adjusted in the cases where it was indicated by the reviewer (TDF).
  • The size of the text in Figure 1 seems appropriate to us and in any case the journal has the original version of this figure to improve the image in the final version of the article.
  • Error bars must be shifted from one another, if they overlap it is hard to assign them to their symbol”.

As indicated in the data analysis, the overlapping confidence intervals is the way to visualize statistical differences between the variance factors. The results between the Pitfall and Winlker traps were very similar to each other (mainly in the rainy season), which produced the result of high overlapping in CIs for these two types of samples. However, this does not limit the ability to make a visual comparison to detect differences between sites and climatic seasons. Precisely the high overlap allows us to quickly rule out that there are no significant differences between the variation factors, in relation to the diversity values.